

# Rosiglitazone accelerates wound healing by improving endothelial precursor cell function and angiogenesis in *db/db* mice

Guoliang Zhou[1,*], Xue Han[2,*], Zhiheng Wu[3], Qiaojuan Shi[2] and Xiaogang Bao[4]

[1] Department of Pharmacy, School of Life and Health Sciences, Anhui Science and Technology University, Fengyang, Anhui, China
[2] Laboratory Animal Center, Zhejiang Academy of Medical Sciences, Hangzhou, Zhejiang Province, China
[3] School of Clinical Medicine, Wannan Medicial Colledge, Wuhu, Anhui, China
[4] Department of Orthopedic Surgery, Spine Center, Changzheng Hospital, Second Military Medical University, Shanghai, China
[*] These authors contributed equally to this work.

Corresponding authors
Qiaojuan Shi, shiqiaojuan@163.com
Xiaogang Bao,
bxg1832178@smmu.edu.cn,
bxg535589@163.com

## ABSTRACT

**Background & Aims.** Endothelial precursor cell (EPC) dysfunction is one of the risk factors for diabetes mellitus (DM) which results in delayed wound healing. Rosiglitazone (RSG) is a frequently prescribed oral glucose-lowering drug. Previous studies have shown the positive effects of RSG on ameliorating EPC dysfunction in diabetic patients. Interestingly, knowledge about RSG with regard to the wound healing process caused by DM is scarce. Therefore, in this study, we investigated the possible actions of RSG on wound healing and the related mechanisms involved in *db/db* diabetic mice.

**Methods.** *Db/db* mice with spontaneous glucose metabolic disorder were used as a type 2 DM model. RSG (20 mg/kg/d, i.g.,) was administered for 4 weeks before wound creation and bone marrow derived EPC (BM-EPC) isolation. Wound closure was assessed by wound area and CD31 staining. Tubule formation and migration assays were used to judge the function of the BM-EPCs. The level of vascular endothelial growth factor (VEGF), stromal cell derived factor-1α (SDF-1α) and insulin signaling was determined by ELISA. Cell viability of the BM-EPCs was measured by CCK-8 assay.

**Results.** RSG significantly accelerated wound healing and improved angiogenesis in *db/db* mice. Bioactivities of tube formation and migration were decreased in *db/db* mice but were elevated by RSG. Level of both VEGF and SDF-1α was increased by RSG in the BM-EPCs of *db/db* mice. Insulin signaling was elevated by RSG reflected in the phosphorylated-to-total AKT in the BM-EPCs. In vitro, RSG improved impaired cell viability and tube formation of BM-EPCs induced by high glucose, but this was prevented by the VEGF inhibitor avastin.

**Conclusion.** Our data demonstrates that RSG has benefits for wound healing and angiogenesis in diabetic mice, and was partially associated with improvement of EPC function through activation of VEGF and stimulation of SDF-1α in *db/db* mice.

## INTRODUCTION

Diabetes mellitus (DM) can cause serious complications, including cardiovascular and cerebrovascular disease, kidney disorders, and diabetic foot ulcers (DFUs; (*Wan et al., 2019*). With the worldwide prevalence of DM and increasing life-spans of diabetic patients, the number of individuals suffering from DFUs has dramatically increased (*Evran Olgun et al., 2019*). Previous studies have shown that the morbidity of foot ulceration in a DM cohort is approximately 15% to 20% in a person's lifetime (*Abbott et al., 2002*), and the average annual cost of this refractory disease is $8,659 per patient (*Tennvall & Apelqvist, 2004*). The financial healthcare burden of DFUs is therefore considerable.

It is widely accepted that delayed wound healing resulting from peripheral circulatory disorders is the critical mechanism in the development of diabetic ulcers (*Forsythe & Hinchliffe, 2016*). Wound healing occurs as a physiologic response to injury and involves neovascularization and angiogenesis (*Hugo et al., 2016*). Endothelial precursor cells (EPCs) have been found to play a remarkable part in the process of neovascularization and maintenance of endothelium homeostasis (*Thal et al., 2012*). These cells can be mobilized from the bone marrow and then participate in angiogenesis triggered by ischemia in a wound (*Wang et al., 2018*). Clinical evidence suggests that EPC transplantation has beneficial effects on wound rehabilitation in patients with ischemic limbs (*Lara-Hernandez et al., 2010*). Consequently, EPCs are a key part of the optimal curative response, but both the number and function of EPCs are impaired in diabetic patients (*Wils, Favre & Bellien, 2017*). Allotransplantation of EPCs from healthy donors also faces the problem of immunological rejections (*Tan et al., 2017*) and the elevation of autologous EPC quantity and/or activity is therefore considered to be a promising therapeutic strategy for angiogenesis-related diseases.

Rosiglitazone (RSG) is a kind of peroxisome proliferator-activated receptor-gamma (PPAR-γ) agonist, commonly used to treat type 2 diabetes mellitus (T2DM), which could enhance insulin sensitivity (*Hiatt, Kaul & Smith, 2013*). Several beneficial effects of RSG on EPCs have been convincingly described. In vitro findings support RSG's attenuation of EPC dysfunction and apoptosis, and the related signaling pathways are involved in ERK/MAPK, NF-κB, and Akt-eNOS of EPCs (*Shengjie et al., 2011*; *Chun et al., 2009*). In addition, clinical research suggests that RSG treatment for 12 weeks improves EPC amounts and migratory ability in patients with T2DM (*Pistrosch et al., 2005*). Nevertheless, the possible impact of PPAR-γ agonist RSG on EPC behavior in wound healing in animals has not yet been described.

On the basis of existing evidence, we hypothesize that the beneficial effects of RSG extend to EPC-mediated wound healing. To test this hypothesis, the present research focuses on determining the influence of RSG on diabetes-induced wound healing and EPC function in genetically diabetic *db/db* mice.

## MATERIALS & METHODS

### Animals

Six-week-old male C57BLKS/J *db/db* mice and male C57BL/6J mice were purchased from the Laboratory Animal Center of Zhejiang Province (Hangzhou, China) with experimental animal use license SYXK 2014-0008. All mice were housed in a 12 h light/dark cycle with an ambient temperature of 24 ± 2 °C and 50% humidity conditions. The mice had free access to water and standard chow. Animals used in this study received humane care in compliance with the National Institutes of Health Guide for the Care and Use of Laboratory Animals. The experimental protocols were approved by the Ethics Committee of Laboratory Animal Care and Welfare, Zhejiang Academy Medical Sciences, with the proved number 2018-141.

### Experimental protocols

Male *db/db* mice with hyperglycemia were used as a T2DM animal model, and all mice began blood glucose monitoring at 8-week-old for 3 weeks to determine diabetes. Random blood glucose was determined with whole blood samples collected from the tail veins by a monitoring system (AB-101G, Maochang, Taiwan). Mice with random blood glucose greater than 300 mg/dL were recognized as diabetes. A total of 32 diabetic mice were randomly divided into two groups at 11-week-old and treated with either RSG (16 mice, 20 mg/kg/d, *i.g.*; *Wang et al., 2017*) or vehicle (16 mice, 0.5% CMC-Na, *i.g.*) for 28 days. The C57BL/6 mice were also randomly divided into a vehicle group (16 mice) and an RSG group (16 mice). At 15-week-old, 10 mice from each group were used for the wound closure experiment and six mice were anesthetized to harvest bone marrow-EPCs (BM-EPCs) (Fig. 1A).

### Evaluation of wound healing

Mice were anesthetized by intraperitoneal injection of ketamine (100 mg/kg). After being fixed on a bubble board and having hair removed from the dorsum, a six mm circle wound was made by punch biopsy (*Han et al., 2017a*; *Han et al., 2017b*). The process of wound healing was monitored by pictures every 2 days until the wounds of the control mice completely healed. The areas of the wounds were evaluated by Image-Pro Plus software version 6.0 (Media Cybernetics, MD, USA) and the rate of wound closure was calculated.

### Quantification of wound angiogenesis

Quantification of wound angiogenesis was conducted by CD31 (a platelet endothelial cell adhesion molecule) immunochemistry and hematoxylin staining (*Han et al., 2017a*; *Han et al., 2017b*). Skin samples following the wound edges were harvested from the experimental mice on days 7 and 14. Collection samples were fixed in 4% paraformaldehyde for 10 h at room temperature and then embedded in paraffin for immunochemistry processing. Samples were cut into 5 µm thick sections, blocked with 5% serum (Chemicon International, Inc., CA, USA) for 3 h, and then incubated with an anti-CD31 antibody (1:500, cat. No. 550274; BD Biosciences, CA, USA) for 1 h. The slides were subsequently incubated with a biotinylated secondary antibody (1:800, cat. No. BA-9200; Vector Laboratories Ltd., Peterborough, UK) for 1 h. Finally, the slides were counterstained

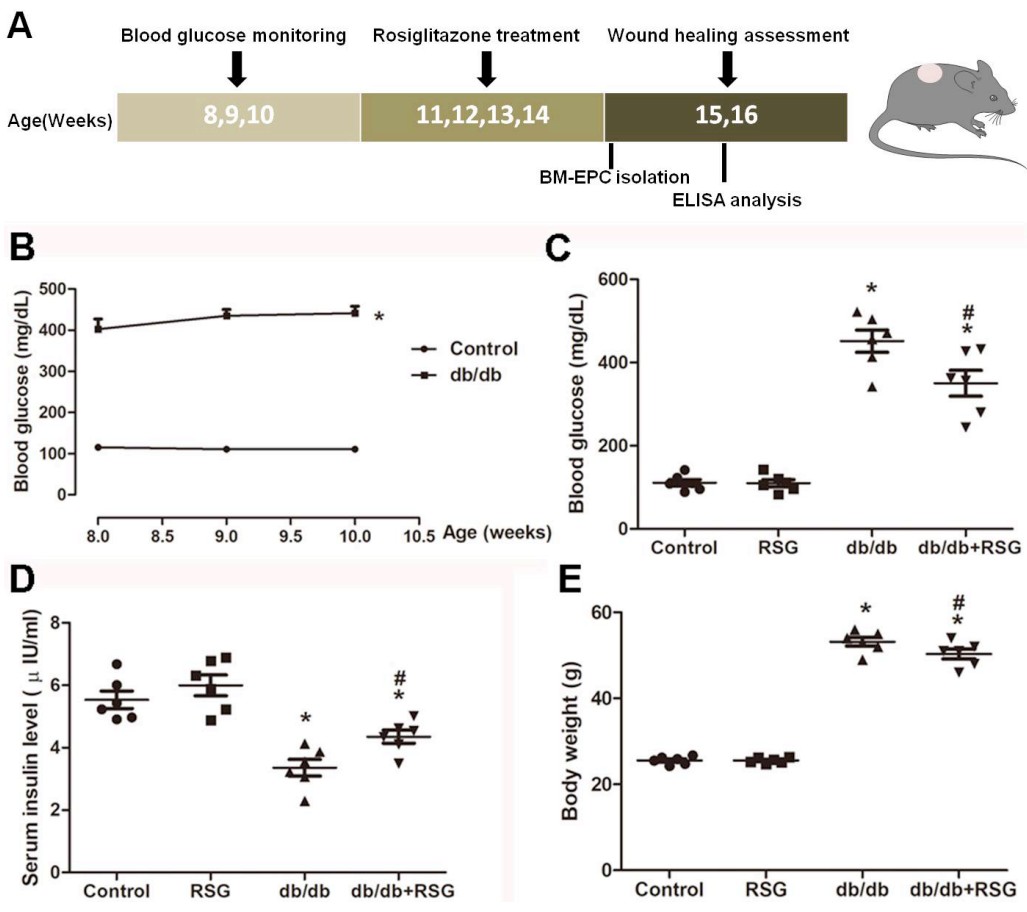

**Figure 1** **Illustration of experimental schedule and establishment of *db/db* diabetic mice model.**
(A) All mice at 8-week-old began blood glucose monitoring for 3 weeks to determine diabetes. *Db/db*
mice were treated with rosiglitazone (20 mg/kg/d × 28d, *i.g.*) or vehicle at 11-week-old for consecutive 4
weeks, and C57BL/6 mice also received RSG or vehicle. At 15-week-old, mice were further divided into 2
cohorts: wound closure creation and BM-EPC harvest. (B) Significant elevation of the blood glucose was
shown in *db/db* mice compared with control group. RSG treatment greatly decreased the blood glucose
(C), increased the serum insulin levels (D) and lighten the body weight in *db/db* mice (E). Values are
mean ± SEM, ($n = 6$ per group). $^*P < 0.05$ *vs.* Control; $^\#P < 0.05$ *vs. db/db*. RSG, rosiglitazone.

with hematoxylin for 2 min. The tubular structure that was positive for CD31 was regarded
as capillary, and the capillary numbers were analyzed. Three sections of slides from each
experimental mouse were evaluated under high-power fields (magnification × 200) by a
light microscope (Leica, Wezlar, Germany).

## Isolation of the BM-EPCs

The BM-EPCs of the mice were isolated and cultured as described previously (*Xie et al.,
2010*). BM-EPCs were extracted from the tibias and femurs of mice and suspended in
endothelial growth medium-2 (Cambrex Corp., East Rutherford, NJ, USA), containing
15% fetal bovine serum (Gibco, Thermo Fisher Scientific, Inc.). Cells were plated into
6-well plates (Coring, MA, USA) at 37 °C with 5% carbon dioxide ($CO_2$). After the cells

were adhered to the plates, non-adherent cells were removed and fresh medium was added to the adhered cells and allowed to incubate for 3 days. BM-EPCs were used for function analysis and the supernatant for ELISA quantification.

## Analysis of BM-EPC function

Tube formation assay was performed to assess BM-EPC function as described previously (*Yu et al., 2016*). Before starting the experiment, a 96-well plate was pre-treated with 50 μl/well of growth factor induced Matrigel (BD Biosciences) for 30 min. Cells were seeded in the pre-treated plate at a concentration of $5 \times 10^5$/ml. After 6 h of incubation, images of the formation tubes were captured under high-power fields (magnification ×50) using a light microscope. The numbers of tubes were then calculated.

The migratory ability assay was also used to evaluate EPC function as previously described (*Yu et al., 2016*). A 24-well transwell plate with modified Boyden chamber (pore size 8 μm; Coring Transwell, Lowell, MA, USA) was used in this assay. A total of 50,000 cells were plated to the upper chamber with 8 μm pores on the polycarbonate membrane, and vascular endothelial growth factor (VEGF; 50 ng/ml; cat. No. V4512; Sigma-Aldrich, Darmstadt, Germany) was added to the medium in the lower chamber. After incubation at 37 °C with 5% $CO_2$ for 24 h, the cells were fixed in 2% paraformaldehyde for 15 min and stained using Hoechst 33258 (10 μg/ml; cat. No. C1011; Beyotime, Shanghai, China). The migration of cells from the upper chamber to the lower was measured under a fluorescence microscope. The number of migrating EPCs was counted.

## Determination of VEGF protein

The supernatant of the BM-EPCs in *db/db* mice after treatment with RSG was collected and the concentrations of VEGF were detected by an ELISA Kit (R&D systems, MN, USA) according to the manufacturer's instruction.

## Detection of stromal cell derived factor -1α (SDF-1α) protein

Blood was harvested from *db/db* mice; the serum concentrations of SDF-1α protein were detected using an ELISA Kit (R&D systems, Minneapolis, MN, USA). Measurements were performed according to the manufacturer's instruction.

## Assessment of BM-EPC insulin signaling

AKT activity was used to evaluate insulin signaling according to previously describe (*Desouza et al., 2011*). BM-EPCs from the C57BL/6 mice and the *db/db* mice were seeded onto the 96-well plate. The cells were treated with or without 1 μmol/L insulin for 1 h to induce the insulin signaling pathway. Then, the phosphorylated AKT and the total AKT of cells was determined by the ELISA Kit (R&D systems, MN, USA) according to manufacturer's instruction. Phosphorylated-to-total AKT ratios were calculated.

## In Vitro Assay

BM-EPCs were obtained from the C57BL/6 mice and cultured *in vitro*. After seven days of cultivation, the medium was replaced with high glucose (HG; 33 mM) medium, HG medium containing RSG (10 nM), or HG medium containing both RSG (10 nM) and avastin (a VEGF inhibitor; 25 μg/ml; (*Merz et al., 2018*) for 24 h. The effects of RSG on

HG-induced EPC activity and tube formation function were determined. Activation of BM-EPCs was determined by cell counting kit -8 (CCK-8; Dojindo Laboratory, Kumamoto, Japan) according to the manufacturer's instruction. EPCs were washed with phosphate buffer (PBS) and then incubated with 10 µl CCK-8 solution for 2 h at 37 °C. The optical density was measured at 450 nm using a Microplate Reader (Bio-Rad, Hercules, CA, USA).

## Statistical analysis

All data is presented as mean $\pm$ standard error of the mean (SEM). Statistical significance was determined with one-way analysis of variance (ANOVA) followed by the Newman-Keuls multiple comparison test of using GraphPad Prism Software version 6. A value for $P$ less than 0.05 was considered to be a statistically significant difference.

# RESULTS

## Effect of rosiglitazone on blood glucose, serum insulin level, and body weight in db/db mice

In *db/db* diabetic mice, the blood glucose level was greatly elevated when compared to the control group (426.4 $\pm$ 1.5 *vs* 112.1 $\pm$ 1.5 mg/dL, $P < 0.05$; Fig. 1B). RSG therapy for 4 weeks significantly decreased the blood glucose concentration and increased serum insulin level compared with *db/db* mice (Fig. 1C: 350.2 $\pm$ 31.1 *vs* 451.6 $\pm$ 26.8 mg/dL, $P < 0.05$; Fig. 1D: 4.3 $\pm$ 0.2 *vs* 3.5 $\pm$ 0.3 µIU/ml, $P < 0.05$). In accordance with the blood glucose improvement, there was also a significant decrease in body weight in the RSG pre-treatment *db/db* mice (Fig. 1E).

## Rosiglitazone accelerated wound closure and wound angiogenesis in db/db mice

To examine the role of RSG treatment on wound healing in *db/db* diabetic mice, the wound area was measured on alternate days until day 14. A significant slowing of wound healing was shown in *db/db* mice when compared to the control group, and RSG greatly accelerated the wound healing in diabetic mice ($P < 0.05$; Figs. 2A–2U). On day 14 post injury, histological assessment showed that the RSG-treated *db/db* wounds had thicker neo-epidermal sheets and more granulation formation than *db/db* diabetic mice (Figs. 3A–3B).

To further assess the effect of RSG on neovascularization, the number of CD31-positive stains indicating the tubular structures, was counted in the wound area and surrounding skin. Figs. 3C–3J show the deteriorated capillary formation on days 7 and 14, respectively, in *db/db* mice, compared with the control group. RSG significantly alleviated capillary formation on both day 7 and day 14 in *db/db* mice ($P < 0.05$; Figs. 3C–3J) suggesting that RSG therapy accelerates wound healing and enhances wound vascularity in diabetic mice.

*Rosiglitazone alleviated BM-EPC function in db/db mice.* Next, we explored the mechanism of RSG in accelerating wound healing using the tube formation assay and migration assay. Impaired BM-EPC function was seen in the diabetic mice when compared to the control group. As expected, administration of RSG attenuated the decline of BM-EPC function in *db/db* mice, reflected by tube formation (21.0 $\pm$ 1.5 *vs* 12.7 $\pm$ 1.1 $P < 0.05$; Figs. 4A–4D, 4I) and migration ability (55.3% $\pm$ 2.5% *vs* 36.2% $\pm$ 3.3% $P < 0.05$; Figs. 4E–4H, 4J).

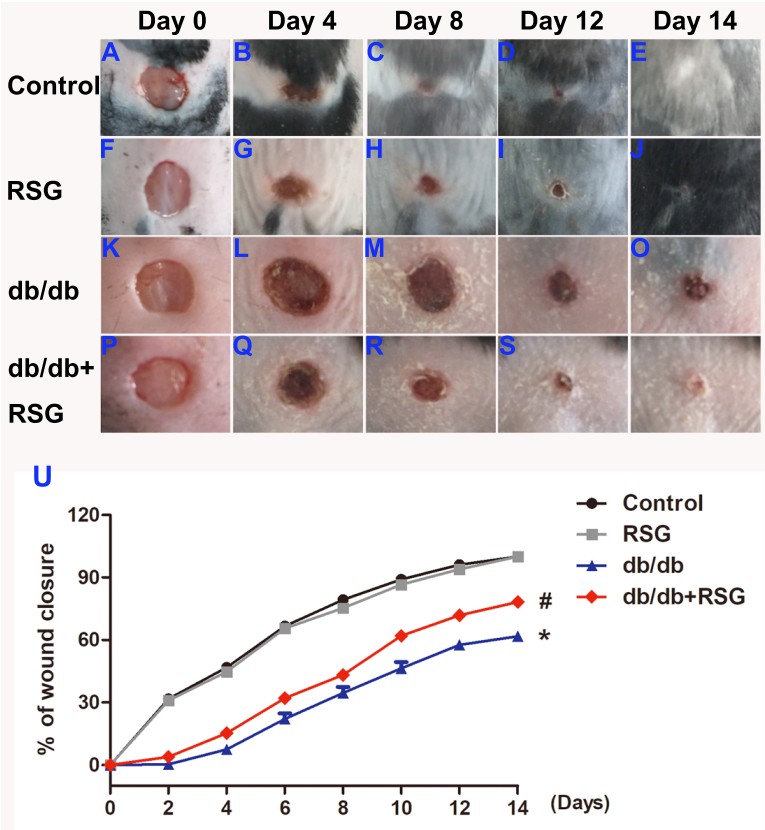

**Figure 2    Rosiglitazone administration accelerated wound closure in *db/db* diabetic mice.** After 4 weeks treated with RSG, *db/db* mice were produced a six mm circle wound on the dorsa by a punch biopsy and then pictures of the wound were taken every 2 days until day 14. RSG treatment accelerated wound closure in *db/db* mice. Representative pictures of wound closure on day 0, day 4, day 8, day12 and day 14 (A–T) and quantitative analysis (U). Values are mean ±SEM, ($n$ = 5 per group). *$P$ < 0.05 *vs.* Control; #$P$ < 0.05 *vs. db/db*. RSG, rosiglitazone.

## Rosiglitazone elevated the level of SDF-1α and VEGF protein in db/db mice

Reduced SDF-1α and VEGF levels have been shown to be involved in serum and impaired homing of EPCs. The levels of SDF-1α and VEGF protein were therefore measured in the diabetic mice. It was found that the level of SDF-1α was significantly decreased in *db/db* mice compared with the control group (2.6 ± 0.3 *vs* 4.6 ± 0.3 ng/mL $P$ < 0.05; Fig. 5A). RSG-treated mice exhibited markedly increased levels of SDF-1α (3.4 ± 0.1 *vs* 2.6 ± 0.3 ng/mL, $P$ < 0.05; Fig. 5A). VEGF levels were significantly lower in the *db/db* mice than in the control group (423.7 ± 19.6 *vs* 614.0 ± 27.1 pg/mL $P$ < 0.05; Fig. 5B), and RSG treatment greatly prevented this decrease (497.3 ± 16.5 *vs* 423.7 ± 19.6 pg/mL $P$ < 0.05; Fig. 5B).

## Rosiglitazone increased insulin signaling in BM-EPCs of db/db mice

Clinical researches have reported that dysfunction of EPCs is positively correlated with insulin resistance in patients with T2DM (*Dei Cas et al., 2011*; *Cubbon, Rajwani*

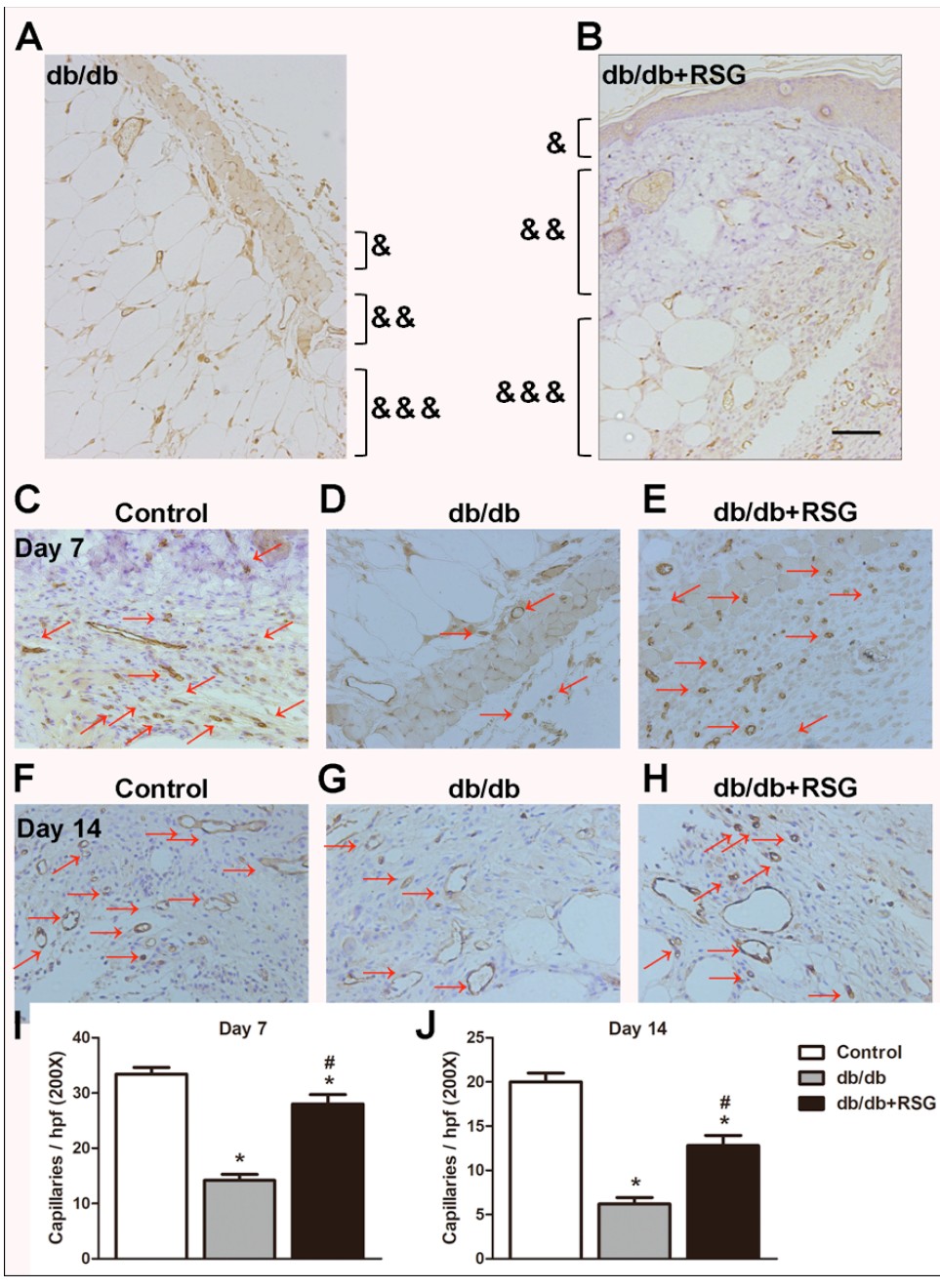

**Figure 3** **Rosiglitazone administration increased wound angiogenesis in *db/db* diabetic mice.** In *db/db* mice, after wound closures were made, skins around the wound were collected and angiogenesis was evaluated on days 7 and 14. Representative pictures of tissue destruction of wounds (A, B) and CD31 staining (C-H). Quantitative study on day 7 (I) and day 14 (J). Red arrows point out CD31-positive capillaries (200 ×; scale bar, 50 μm). Values are mean ±SEM, ($n = 5$ per group). $^*P < 0.05$ *vs.* Control; $^\#P < 0.05$ *vs. db/db*. (&) Epidermis, (&&) dermis and (&&&) subcutis indicated. RSG, rosiglitazone. hpf, high power field.

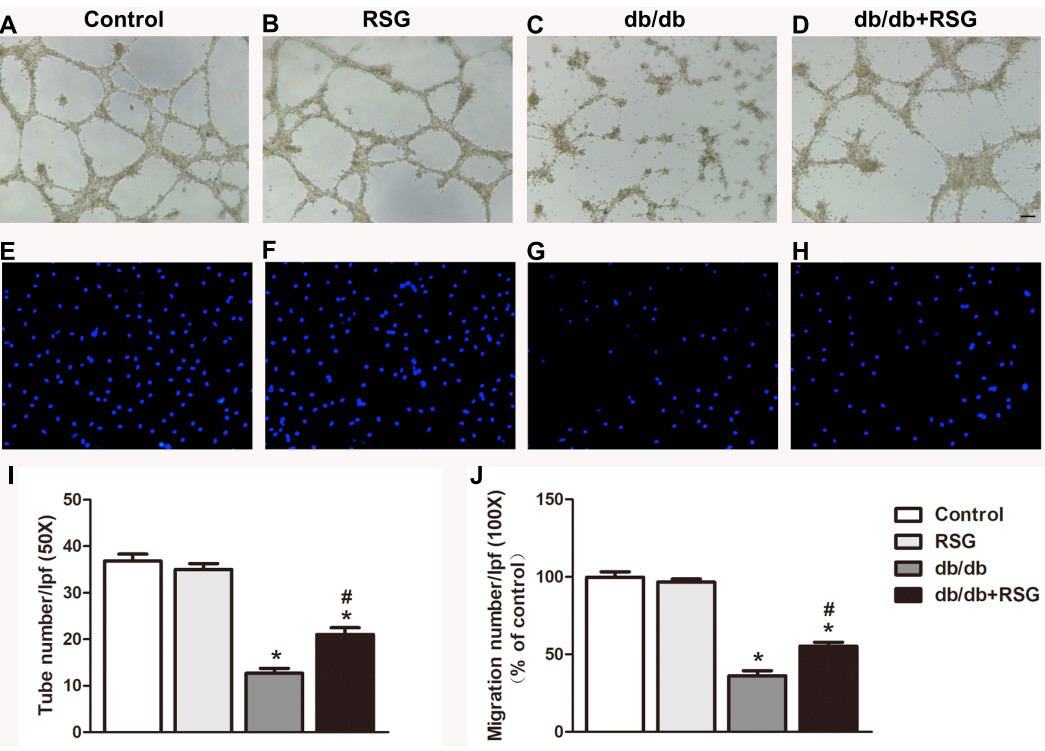

**Figure 4** **Rosiglitazone administration improved BM-EPC function in *db/db* diabetic mice.** In *db/db* mice, BM-EPCs were isolated and cultured after 4 weeks RSG therapy. BM-EPC function was estimated by tube formation assay and migration assay. RSG treatment elevated tube formation ability (A–D, I) and migration ability (E–H, J) of BM-EPCs. (A) 50×; scale bar, 100 μm; B: 100×; scale bar, 100 μm. Values are mean ± SEM, ($n = 6$ per group). *$P < 0.05$ *vs.* Control; #$P < 0.05$ *vs. db/db*. RSG, rosiglitazone. lpf, low power field.

*& Wheatcroft, 2007*). Here we determined whether DM environment induces insulin signaling defects in EPCs and whether treatment with RSG reduces the defects. To test this hypothesis, we used BM-EPCs stimulated by insulin and measured AKT activity. As illustrated in Fig. 5C, BM-EPCs obtained from the *db/db* mice were markedly insulin resistant as measured by phosphorylated-to-total AKT ratio compared with BM-EPCs from the C57BL/6 mice ($2.2 \pm 0.04$ *vs* $5.6 \pm 0.2$ $P < 0.05$). AKT activity was significantly up-regulated in BM-EPCs from RSG-treated *db/db* mice when compared with the vehicle treated *db/db* mice ($4.2 \pm 0.3$ *vs* $2.2 \pm 0.04$ $P < 0.05$; Fig. 5C).

## Avastin prevented the action of RSG in EPCs in vitro

High glucose is commonly used as an EPC injury model *in vitro*. To clarify the role of VEGF in EPC function induced by high glucose, avastin a VEGF inhibitor was used. The cell viability and capacity for tube formation of BM-EPCs were greatly impaired by high glucose. RSG (10 nM) treatment significantly increased cell viability ($85.4 \pm 1.9$ *vs* $73.4 \pm 2.7\%$, $P < 0.05$; Fig. 6A) and improved the impaired EPC function ($30.2 \pm 1.4$ *vs* $19.8 \pm 1.5$, $P < 0.05$; Figs. 6B–6F). Inhibition of VEGF levels abolished the enhanced EPC

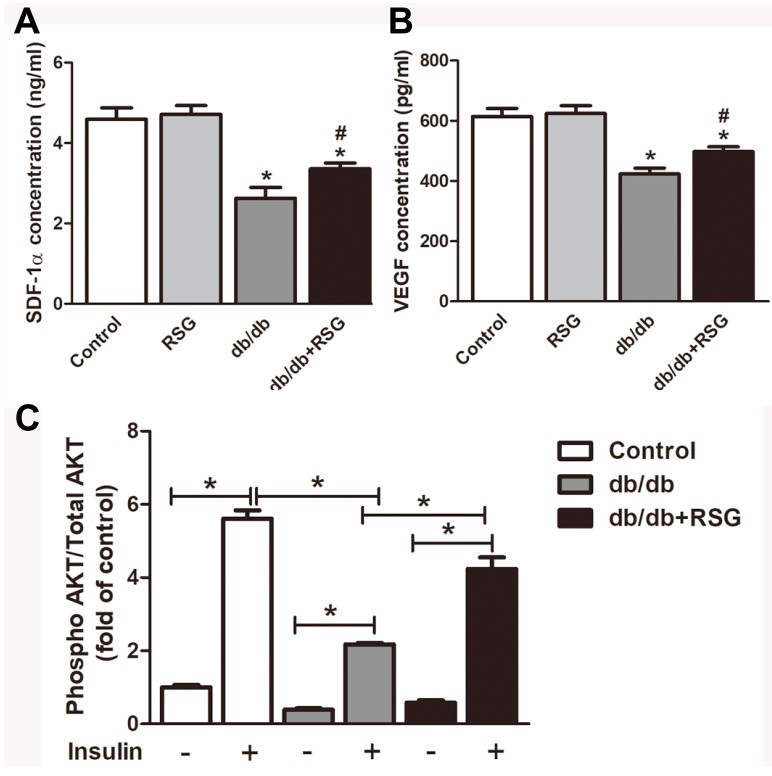

**Figure 5** **Rosiglitazone administration increases the levels of VEGF and SDF-1α protein and improved insulin resistant in _db/db_ mice.** (A) Effect of RSG treatment on SDF-1α protein levels in serum of the _db/db_ mice. (B) ELISA analyses of VEGF levels in supernatant of BM-EPCs. (C) BM-EPCs treated with or without 1 μmol/L insulin for 1 h, and concentrations of phosphorylated AKT and total AKT were determined by the ELISA kit. Values are mean ± SEM, (A, B: _n_ = 6; C: _n_ = 5 per group). A, B: *_P_ < 0.05 _vs._ Control; A, B: #_P_ < 0.05 _vs. db/db_. RSG, rosiglitazone.

viability and function mediated by RSG (EPC viability: 73.3 ± 2.6 _vs_ 85.4 ± 1.9%, _P_ < 0.05; Fig. 6A; tube number: 22.0 ± 1.7 _vs_ 30.2 ± 1.4, _P_ < 0.05; Figs. 6B–6F).

## DISCUSSION

Three principal findings arose from this study. First, administration of RSG accelerated wound healing and improved angiogenesis in _db/db_ diabetic mice. Second, we found that RSG alleviated the impaired BM-EPC capacity and insulin resistance in _db/db_ mice. Third, we showed that RSG elevated VEGF and SDF-1α levels in diabetic mice, and inhibition of VEGF could abolish the improved EPC viability and tube formation function mediated by RSG _in vitro_.

D_b/db_ mice have become a well-established animal model for T2DM that has been widely used in animal experiments. As a consequence of gene mutation of the leptin receptor, _db/db_ mice exhibit spontaneous glucose metabolic disorder, which is similar to the clinical symptoms of adult-onset T2DM. There is strong evidence that suggests that sustained hyperglycemia cannot occur until 8 weeks old (_Bao et al., 2016_), so 8-week-old mice were therefore selected for this study, and stable high blood glucose was shown from

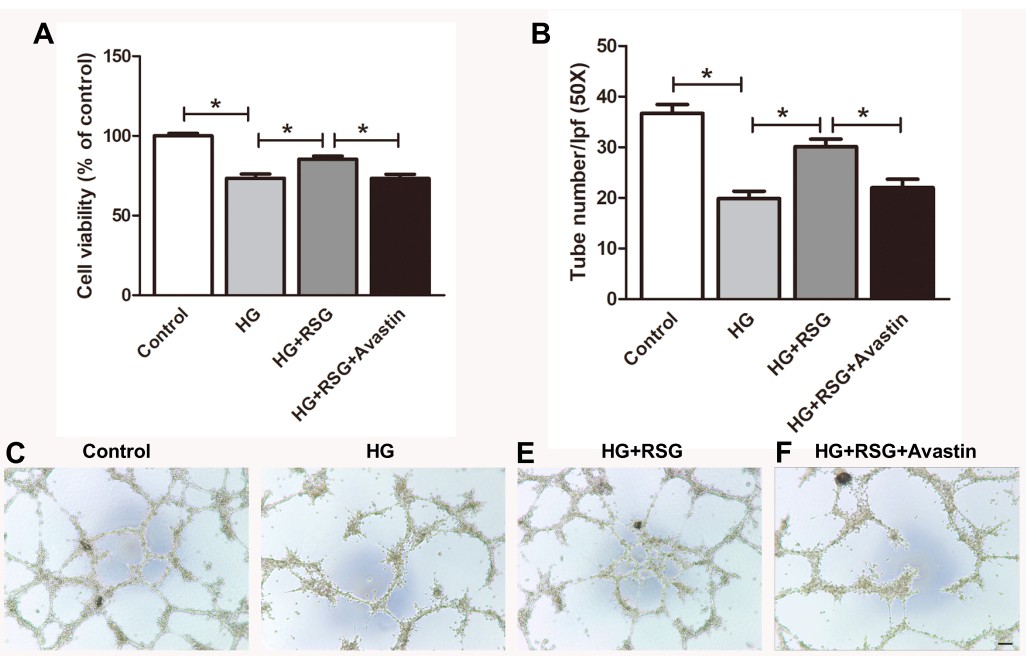

**Figure 6** **VEGF inhibitor avastin prevented the role of RSG in BM-EPCs induced by high glucose.** (A) BM-EPCs exposed to 33 mmol/L glucose with or without RSG (10 nM) and VEGF inhibitor Avastin (25 μg/ml) for 24 h. CCK-8 detected EPC viability *in vitro*. (B–F) Avastin abolished the elevated tube formation ability by RSG (C–F: 50×; scale bar, 100μm). Values are mean ± SEM, ($n = 6$ per group). RSG, rosiglitazone. HG, high glucose.

week 8 to week 10 (Fig. 1B). The results suggest that a typically diabetic animal model was established.

Wound healing following tissue injury occurs as a cellular response, which involves activation of endothelial cells and the release of numerous signaling molecules (*Longmate et al., 2017*). Individuals with DM and associated high blood glucose suffer wound healing illness and subsequent complications, such as peripheral vascular disorder and lower limb amputation (*Brem & Tomic-Canic, 2007*). It has been long understood that angiogenesis plays an important role in the process of wound healing at many stages. Physiologically, angiogenesis takes part in the formation of new blood vessels, and this is involved in endothelial cell repair and especially BM-EPC mobilization (*Saboo et al., 2016*). CD31, also called platelet endothelial cell adhesion molecule -1 (PECAM-1), is a marker that indicates existing endothelial cells and vascular formation in immunohistochemical assay (*Lertkiatmongkol et al., 2016*). Consistent with previous studies (*Han et al., 2017a*; *Han et al., 2017b*), we found that angiogenesis capacity at the wound sites, as indicated by CD31 staining, was impaired in the *db/db* diabetic mice compared to the non-diabetic subjects. However, the underlying mechanism of angiogenesis under the pathologic condition of DM remains a significant aspect yet to be explained.

EPCs have previously been reported to have a close relationship with neovascularization in response to complicated diabetic wound. EPCs in bone marrow respond to ischemia

with chemokine factors, which result in the homing of the impaired site, where they are implicated in vascular repair and new vessel formation as mature endothelial cells (*Gao et al., 2008*). Unexpectedly, studies have indicated that the mobilization ability of EPCs from bone marrow to peripheral blood declines much more in diabetic patients than in normal individuals, resulting in decreased numbers of circulating EPCs (*Falay & Aktas, 2016*). In line with these results, we have found that the activity and total numbers of EPCs were reduced both in *db/db* mice and in mice exposed to streptozocin (*Han et al., 2017a*; *Han et al., 2017b*; *Yu et al., 2016*). RSG, a PPAR-γ agonist family member, is approved for use as a single drug or in combination with metformin therapy for T2DM. The hypoglycemic mechanism of RSG is recognized as a mediator of increased insulin sensitivity in muscles and adipose tissues (*Kolak et al., 2007*). Previous studies have reported that RSG could reduce EPC apoptosis and improve impaired EPC-induced angiogenesis, suggesting that RSG has beneficial effects on EPC biology (*Verma et al., 2004*; *Haberzettl et al., 2016*). Clinical studies have also demonstrated that RSG contributes to improved EPC numbers and migratory function in T2DM patients (*Pistrosch et al., 2005*), but the associated mechanisms have been largely unclear. In the current study, we found that administration of RSG accelerated wound closure and increased capillary density, accompanied by improvement of the tube formation and migratory activity of EPCs in *db/db* mice. These results imply that the efficacy of RSG in wound healing was potentially related to regulation of EPC dysfunction under diabetic pathology conditions. Furthermore, in human, dysfunction of EPCs is also in the presence of the insulin resistance syndrome (*Desouza et al., 2011*). Our results showed that RSG ameliorated insulin signaling defects in BM-EPCs of *db/db* mice.

VEGF has been shown to be a pivotal player in promoting endothelial cell migration (*Morales-Ruiz et al., 2000*; *Dimmeler, Dernbach & Zeiher, 2000*) and EPC-mediated angiogenesis. Blood VEGF levels are indeed positively associated with good prognosis for wounds in diabetes. For RSG, there is controversial data regarding the influence on VEGF expression. On one hand, *Ku et al. (2017)* reported that RSG might promote neovascularization and vascular permeability by inducing VEGF expression, and myofibroblasts exhibit an up-regulation of VEGF-dependent PPAR-γ inhibition (*Chintalgattu et al., 2007*). On the other hand, RSG failed to increase VEGF level in cultured mesangial cells under high glucose conditions (*Whiteside et al., 2009*) and inhibited biosynthesis of VEGF in keratinocytes (*Schiefelbein et al., 2008*). In the current study, RSG increased VEGF expression in BM-EPCs of diabetic mice. Inhibition of VEGF abolished the enhanced EPC viability and tube formation mediated by RSG. Furthermore, EPC recruitment to the site of the wound depends on up-regulation of the SDF-1α. Preclinical studies have demonstrated a decrease in SDF-1α at wound sites in diabetic mice and this decrease is responsible for reduced EPC homing (*Gallagher et al., 2007*). The present study determined that treatment with RSG significantly up-regulated SDF-1α levels accompanying accelerated wound healing in *db/db* mice.

Actually, we did have some limitations for this study. In particular, the mice dorsal wounds exhibit furious contraction during healing, which is different from the process in human wounds, where heal is through granulation and reepithelialization (*Konop et al., 2017*). We did not use the wound splinting model to prevent skin contraction

(*Konop et al., 2017*; *Wang et al., 2013*), even our wound model was used as previously reported (*Han et al., 2017a*; *Han et al., 2017b*; *Yu et al., 2016*; *Li et al., 2016*).

## CONCLUSION

We demonstrated that RSG therapy protects against diabetes-induced wound healing delay, angiogenesis disorder, and EPC dysfunction in *db/db* diabetic mice. Administration of RSG accelerated wound closure, which was partially correlated with an amelioration of EPC function through activation of VEGF/SDF-1α and insulin signaling. These results suggest that RSG may be a promising drug for local wound closure of DFUs, given its well-established pharmacological action of glucose-lowering.

### Funding

This study was supported by the Natural Science Foundation of Zhejiang (Grants LGJ18H310002 and LQY19H090001), National Natural Science Foundation of China (No. 31900381 and 31870080), the Medical Scientific Research Foundation of Zhejiang Province (No. 2020388156), the Youth start-up fund of Second Military Medical University (Grant No. 2018QN13), and the Innovation Training Program of Anhui (No. 201810368117). The funders had no role in study design, data collection and analysis, decision to publish, or preparation of the manuscript.

### Grant Disclosures

The following grant information was disclosed by the authors:
Natural Science Foundation of Zhejiang: LGJ18H310002, LQY19H090001.
National Natural Science Foundation of China: 31900381, 31870080.
Medical Scientific Research Foundation of Zhejiang Province: 2020388156.
Youth start-up fund of Second Military Medical University: 2018QN13.
Innovation Training Program of Anhui: 201810368117.

### Competing Interests

The authors declare there are no competing interests.

### Author Contributions

- Guoliang Zhou performed the experiments, analyzed the data, contributed reagents/materials/analysis tools, prepared figures and/or tables, authored or reviewed drafts of the paper, approved the final draft.
- Xue Han performed the experiments, prepared figures and/or tables.
- Zhiheng Wu performed the experiments.
- Qiaojuan Shi conceived and designed the experiments.
- Xiaogang Bao conceived and designed the experiments, contributed reagents/materials/analysis tools, authored or reviewed drafts of the paper.

## Animal Ethics

The following information was supplied relating to ethical approvals (i.e., approving body and any reference numbers):

The Ethics Committee of Laboratory Animal Care and Welfare, Zhejiang Academy Medical Sciences, provided full approval for this research with proved number (#2018-141). An experimental animals use license (NO: SYXK 2014-0008) of Laboratory Animal Center of Zhejiang Province was obtained.

## Data Availability

The raw data are available in the Supplemental Files.

## Supplemental Information

Supplemental information for this article can be found online at http://dx.doi.org/10.7717/peerj.7815#supplemental-information.

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
