# Peer review of "Rosiglitazone accelerates wound healing by improving endothelial precursor cell function and angiogenesis in db/db mice"

_PeerJ, doi:10.7717/peerj.7815_

## Round 0.1 · original submission · Major Revisions

Dear Dr. Zhou,

Your manuscript entitled " Rosiglitazone accelerates wound healing by improving endothelial precursor cell function in db/db mice" which you submitted to PeerJ, has been reviewed by the editor and 3 experts in the field.

I regret to inform you that the reviewers have raised serious concerns, and therefore your paper cannot be accepted for publication in PeerJ in its present form. However, since reviewers felt the manuscript contained some potentially interesting results, I would be willing to reconsider if you wish to undertake major revisions and resubmit.

If you decide to resubmit the revised version, please summarize all the improvements made in the new version and give answers to all critical points raised in the reviewers’ report in an accompanying letter. Please copy and paste each and every reviewer's comment above your response. If you feel any of their points are inappropriate, you are certainly free to provide rebuttal in your covering letter.

The English language should be improved to ensure that your international audience can clearly understand your text. I suggest that you have a native English speaking colleague review your manuscript. The current phrasing makes comprehension difficult. Please also provide the information requested about the experimental design and include a section that addresses the study's limitations. Technical/protocol details, potential limitations of experimental design etc., must be openly discussed.

Please note that resubmitting your manuscript does not guarantee eventual acceptance. Since the requested changes are major, and also include new experimental work, the revised manuscript will undergo a second round of review by the same reviewers. I must emphasize that the acceptability of the revision will depend upon the resolution of the points raised by the reviewers.

Sincerely yours,

Stefano Menini

Reviewer 1 ·

Basic reporting

1. English writing should be reviewed by a native speaker
2. Legend of some axis should be explained, what does hpf mean? (y axis in figure 3b and c) What does lpf mean? (y axis in figure 4c and d)
3. Figure 2a does not have a scale bar in at least one picture

Experimental design

1. Why a control group with RSG was not included?
2. In what volume was RSG given to animals
3. How many animals were used in each group?
4. How many animals from each group were used for cell extraction from bones?
5. If VEGF was measured by ELISA in figure 5 why the units in Y axis are fold of control instead of concentration?

Validity of the findings

Experiments are well performed but they are mere correlations.
There are no clear conclusions besides a very general one that indicates that RSG is beneficial for the parameters measured

Additional comments

In this manuscript authors evaluate the effect of Rosiglitazone in wound healing in a mouse diabetic model. Upon reading several concerns have rised:
1. How was determined the dose of RSG administered to mice?
2. On lane 245 authors conclude from the experiment that “RSG protects against wound healing injury. Which results are considered to make this conclusion?
3. Lane 252, authors did not measure the expression (synthesis) of the proteins, they measured the abundance of them (that includes processes of synthesis and degradation)
4. A better characterization of BM-EPC is needed, are these cells insulin resistant? Did the treatment with RSG reduce the resistance?
5. Which is the important factor in the results observed by the administration of RSG?, the reduction in glycemia?, the increase in Insulin?
6. If VEGF or SDF-1 are not present, can RSG improve the process of wound healing, and cell proliferation?

·

Basic reporting

A work presented for review entitled "Rosiglitazone accelerates wound healing by improving endothelial precursor cell function in db / db mice" is an interesting contribution to the development of basic and clinical sciences. The work is written in the correct language, however, in a few places, the abbreviations that are introduced should be developed. The structure of the article is correct.

Experimental design

The experimental design is correct, however maybe interesting would be to give the drug to non genetic mice and compare this between group.

I have a few question:

1. Where was the control wound located? Please insert a macroscopic photo of both wounds.
2. The attached photos do not show any stitches on the wound. In what mechanism did the wounds heal? Was it generous by granulation or contraction?
3. Do you measure level of any cytokine. In literature we can find information that RSG showed anti-inflammatory properties (https://www.sciencedirect.com/science/article/pii/S0022202X15371050, https://www.ncbi.nlm.nih.gov/pmc/articles/PMC4834268/)
4. Have you performed histological tests (review of HE staining) or immunohistochemical tissues taken from the wound? I found in the manuscript that only these stains were made to evaluate angiogenesis. Am I right? I recommended to perform this staining and evaluate tissue responses. It could be semi-quantitative measurements. See this papers: DOI: 10.1177/0885328218801114, https://onlinelibrary.wiley.com/doi/abs/10.1111/wrr.12500
DOI: 10.1177/0963689718807410
5. You can also measure the thickens of dermal layer.

Validity of the findings

No comment

Additional comments

My general comments are mentioned above.

Reviewer 3 ·

Basic reporting

I found the manuscript difficult to read on a first pass as the grammar, spelling, and sentence structure needs improvement. I’ve made a few suggestions below, but I would also advise a thorough read through and check before resubmission.
Line 83: remove “the” before angiogenesis. Hypoxia (reduced oxygen) is a consequence of Ischemia (reduced blood flow), you can use either term, but not both as they elude to the same things
Line 91: “is a kind of…” instead of “as a kind of…”
Line 92: “to treat type 2 DM” instead of “to therapy of type 2 DM”
Line 94: “signalling” instead of “signal”
Line 170: “suspended” instead of “slightly dispersed”
Line 171: “plated” instead of “planted”
Line 174: “ELISA” instead of “Elisa”
Line 305: “role” instead of “partly”
Line 312: “underlying” instead of “underline”

Line 93: In this line you refer to “side effects” of RSG on EPC, do you mean the previously found effect of RSG treatment or result of RSG treatment on EPC? Please clarify the use of the phrase “side effect”

Experimental design

In this manuscript you have used a wound protocol cited by Han et al 2017, this protocol does not use silicone splints to keep the wounds from healing by contraction. For comparison to human skin wounds mouse wounds need to be splinted to cause healing by re-epithelialization (as seen in human skin) rather than healing by skin contraction (as seen in mouse skin). Please refer to Dunn et al, J. Vis. Exp. (75), e50265, (2013), and Wang et al Nature Protocols volume 8, pages 302–309 (2013) for protocols on splinting of wounds in mice. Please discuss these papers and why a protocol without splinting was used. This is very important.

I found the explanation for experimental protocol difficult to follow, it would benefit the reader if you break up the explanation into 3 main components ie: 1) all mice (db/db and C57BL/6) at 8 weeks old began blood glucose monitoring for 3 weeks to determine diabetes, 2) treatment with RSG or vehicle commenced at 11 weeks old for 28 days in diabetic mice, C57BL/6 mice received vehicle only, 3) at 15 weeks old, mice were further divided into 2 cohorts (bone marrow harvest and wound closure experiment). It is unclear which mice were used for bone marrow harvest, are they diabetic with RSG treatment, diabetic with vehicle treatment or C57BL/6 with vehicle treatment? Please clarify. The use of graphical timeline, as in figure 1, is beneficial, perhaps a graphic to explain the protocol would also be beneficial.

Line 166, how many sections were on each slide? One slide or one section is quite low for analysis. I suggest analysing a minimum of 3 sections per animal.

Line 171-174, “Cells were planted into 6-well plates (Coring, MA, USA) at 37 OC, 5% carbon dioxide (CO2). After the cells were adhere to plates, a few of non-adherent cells were absorbed and added to the fresh medium. For another 3 days culturing…” This section did not make sense initially, however after looking at the references you provided I believe you wanted to write that “non-adherent cells were removed, and fresh medium was added to the adhered cells and allowed to incubate for 3 days”. Please clarify and fix if incorrect.

Line 143: please detail the type of monitoring system used for blood glucose.

Line 182, for the transwell plate, what pore size and pore material was used?

Validity of the findings

Prolonged inflammation is a key problem in wound healing for diabetic patients. Given the promising effects of Rosiglitazone it would be beneficial to determine if key inflammatory markers involved in wound healing such as CCL2, CXCL8, TGF-β… are reduced in your Rosiglitazone treated mice at both time points

Additional comments

Is it possible to get brighter, higher quality images?

Overall a good paper, needs some refining and clarification to get the correct point across. I recommend completing the suggested experiments to strengthen the manuscript.

---

## Round 0.2 · accepted · Accept

Dear Dr. Bao,

Our referees have now considered your manuscript entitled "Rosiglitazone accelerates wound healing by improving endothelial precursor cell function in db/db mice" and have recommended publication in “PeerJ”. We are pleased to accept your paper in its current form which will now be forwarded to the publisher for typesetting.

I thank all reviewers for their effort in improving the manuscript and the authors for their cooperation throughout the review process.

Yours sincerely,

Stefano Menini

Reviewer 1 ·

Basic reporting

1. Clear and unambiguous, professional English used throughout.
English has been reviewed and corrected
2. Literature references, sufficient field background/context provided.
Sufficient introduction and background has been included, and appropriately referenced.
3. Professional article structure, figures, tables. Raw data shared.
The structure of the article is adequate. Figures are relevant to the content of the article, and of sufficient resolution. Figures are now appropriately described and labeled.
4. All appropriate raw data have been made available

5. Self-contained with relevant results to hypotheses.
The submission is self-contained, represents an appropriate ‘unit of publication’, and includes all results relevant to the hypothesis.

Experimental design

EXPERIMENTAL DESIGN
1. The Original primary research is within Scope of the journal.
2. Research question are well defined, relevant & meaningful.
3. It is stated how the research fills an identified knowledge gap.
4. It is a Rigorous investigation performed to a high technical & ethical standard.
5. Methods now are described with sufficient detail & information to replicate.

Validity of the findings

Methods, results, discussion and conclusions have been revised, complemented and now are clear.

Additional comments

The following is the review for the manuscript entitled “Rosiglitazone accelerates wound healing by improving endothelial precursor cell function in db/db mice”
in this new version all the concerns raised by this reviewer have been answered.
Although the experiment with the antagonist of VEGF, was not done in cells obtained from db/db mice, it demonstrates the participation of VEGF on wound healing in this model. It would be interesting to demonstrate whether a similar effect to the one observed in cells treated with glucose (acute state) is also observed in cells from db/db mice (chronic state) treated with Rosiglitazone.

·

Basic reporting

In my opinion, after making corrections, the manuscript contains the necessary information to correctly understand and replicate the study by another authors.

Experimental design

After correction this paper well described all technical details.

Validity of the findings

The paper presented for review raises very important issues related to the possible use of RSG on wound healing and the related mechanisms involved in db/db diabetic mice.

Reviewer 3 ·

Basic reporting

no comment

Experimental design

no comment

Validity of the findings

no comment

Additional comments

The authors have sufficiently corrected the manuscript based on the reviewers comments